# Hospital load and increased COVID-19 related mortality in Israel

Hagai Rossman [1,2,6], Tomer Meir[1,2,6], Jonathan Somer[3], Smadar Shilo [1,2,4], Rom Gutman[3], Asaf Ben Arie[5], Eran Segal [1,2 ✉], Uri Shalit[3,7 ✉] & Malka Gorfine [5,7 ✉]

The spread of Coronavirus disease 19 (COVID-19) has led to many healthcare systems being overwhelmed by the rapid emergence of new cases. Here, we study the ramifications of hospital load due to COVID-19 morbidity on in-hospital mortality of patients with COVID-19 by analyzing records of all 22,636 COVID-19 patients hospitalized in Israel from mid-July 2020 to mid-January 2021. We show that even under moderately heavy patient load (>500 countrywide hospitalized severely-ill patients; the Israeli Ministry of Health defined 800 severely-ill patients as the maximum capacity allowing adequate treatment), in-hospital mortality rate of patients with COVID-19 significantly increased compared to periods of lower patient load (250–500 severely-ill patients): 14-day mortality rates were 22.1% (Standard Error 3.1%) higher (mid-September to mid-October) and 27.2% (Standard Error 3.3%) higher (mid-December to mid-January). We further show this higher mortality rate cannot be attributed to changes in the patient population during periods of heavier load.

¹ Department of Computer Science and Applied Mathematics, Weizmann Institute of Science, Rehovot, Israel. ² Department of Molecular Cell Biology, Weizmann Institute of Science, Rehovot, Israel. ³ Technion - Israel Institute of Technology, Haifa, Israel. ⁴ Pediatric Diabetes Clinic, Institute of Diabetes, Endocrinology and Metabolism, Rambam Health Care Campus, Haifa, Israel. ⁵ Department of Statistics and Operations Research, Tel Aviv University, Ramat Aviv, Israel. ⁶These authors contributed equally: Hagai Rossman, Tomer Meir. ⁷These authors jointly supervised this work: Uri Shalit, Malka Gorfine. ✉email: eran.segal@weizmann.ac.il; urishalit@technion.ac.il; gorfinem@tauex.tau.ac.il

The rapid spread of severe acute respiratory syndrome coronavirus 2 (SARS-CoV-2) worldwide and the disease caused by the virus, coronavirus disease 19 (COVID-19), has caused a global pandemic[1] with devastating social and economic consequences. Throughout the pandemic, several health systems were overwhelmed in light of the rapid emergence of new cases within a short period of time. Notable examples include Lombardy in Italy[2] in which ICU capacity has reached its limit in March 2020, and New York city in the USA[3]. The heavy workload imposed on hospital services might have negatively affected patients' outcomes and exacerbated mortality rates.

The first individual with COVID-19 in Israel was identified on 21 February 2020. In response, the Israeli Ministry of Health (MOH) has gradually employed a series of social distancing measures in order to mitigate the spread of the virus[4]. Following the relaxation of these measures in May 2020, the number of new patients has substantially increased and on the 10 September 2020, Israel became the country with the highest rate of COVID-19 infections per capita worldwide[5]. This period was referred to as the second wave of COVID-19 disease and as a result, Israel was the first country worldwide to impose a second lockdown in mid September 2020. The restrictions were gradually loosened during October 2020, followed by an additional increase in the number of cases leading the government to impose a third lockdown in January 2021. In-hospital mortality of individuals with COVID-19 throughout the study period is described in Fig. 1a.

Here, we assessed excess mortality using a model developed for predicting patient mortality based on data of day-by-day patient disease course[6]. We show that during a peak of hospitalizations in September and October 2020, patient deaths significantly exceeded the model's mortality predictions, while reverting to match the predictions as patient load subsided in October, and then showing renewed excess mortality as hospital load has increased again since late December. As Israel has a relatively small geographical area and small population, and national restrictions were placed on the entire country at the same time (such as school closures and lockdowns) – cases and hospitalizations dynamics were similar across different hospitals, and so the analysis was conducted on a national level. Hospitalization dynamics for Israel's 15 largest hospitals, and which account for almost 75% of national COVID-19 related hospitalizations during this period, are shown in Supplementary Fig. 1.

## Results

We applied a prediction model on nationwide hospitalization data originating from the Israeli MOH to predict mortality of hospitalized patients with COVID-19 based on their age, sex, and clinical state on their first day of hospitalization. In-hospital mortality predictions were made using Monte-Carlo methods based on a multistate survival analysis (see below) and a set of Cox regression models, first constructed and validated on a nationwide cohort during the first stages of the pandemic in Israel[6] (see Methods section). Overall, from 15/07/2020 to 20/01/2021, 22,636 individuals were hospitalized with COVID-19 infection in Israel and were included in the analysis. Mean age was 59.8 ± 21.9 years old (median age was 63 years old), and 11,070 (48.9%) were females. We first divided the data into 27 weeks, and assigned each patient according to week of hospitalization. Patients' characteristics are presented in Table 1. Patients who died on the same day of hospital admission were not included in the analysis.

The model, already trained and validated on a cohort of 2703 hospitalized patients early in the course of the pandemic[6], was modified, re-trained and validated on data of 5966 individuals,

hospitalized from 15/07/2020 to 08/09/2020 (time-period I). The model was then applied to data of individuals hospitalized from 09/09/2020 to 20/01/2021. We divided this interval into three time periods (denoted II, III, IV) according to hospital load: Periods II (09/09/2020 to 27/10/2020) and IV (15/12/2020 to 20/01/2021) are those where the daily number of severe+critical patients exceeded 500; see Fig. 1b. Excess mortality was calculated as the difference between observed and predicted mortality. Strikingly, during time-periods II and IV, the observed 14-day in-hospital mortality was respectively 22.1% (Standard Error (SE) 3.1%) and 27.2% (SE 3.3%) higher than predicted by the model (Table 2), whereas our model accurately predicted in-hospital mortality during periods I and III, where the observed 14-day mortality was 0.6% (SE 5.1%) and 10.8% (SE 5.7%) higher than predicted by the model respectively; see Fig. 1c and Table 2. Similar trends were observed for 28-day mortality (Table 2). Cumulative expected and actual death curves for each hospitalization week are presented in Fig. 2; Supplementary Fig. 2 presents calibration plots by hospitalization week.

## Discussion

In this study, we show that even under moderately heavy patient load (above 500 severely-ill patients hospitalized nationwide), in-hospital mortality rate of patients with COVID-19 in Israel significantly increased compared to periods of lower patient load (250–500 severely-ill patients). Notably, the threshold defined by policy makers in Israel as an upper bound in which the healthcare system will not be able to adequately treat patients was 800 severe and critical patients[7]. In addition, the increase in observed mortality was evident despite the fact that throughout the pandemic, clinical experience in treatment of COVID-19 patients increased, along with a better understanding in pharmacologic (such as corticosteroids[8] and remdesivir[9]) and non-pharmacologic (such as proning[10]) treatment modalities that may be beneficial for the patients.

We postulate that the excess mortality is most likely due to the rapid escalation in the number of hospitalized patients with COVID-19 during these time periods in Israel, which may have resulted in an insufficiency of health-care resources, thereby negatively affecting patient outcomes. Theoretically, several other explanations may account for the increased mortality observed. First, it is possible that the excess deaths may be driven by a more vulnerable patient population, with a higher risk for mortality, that was admitted for hospitalization around time-periods II and IV. However, the model adjusts for age, sex, and clinical state upon 1st day of hospitalization, and the predictions therefore take these differences into account. Although our data did not include information on patient comorbidities that may also influence the mortality rate from COVID-19, it was previously shown that this information is not necessarily essential for accurate mortality prediction for hospitalized patients when utilizing a multistate survival model which takes into account the patient's clinical state upon hospitalization[6]. Accounting for clinical state also somewhat reduces the likelihood of increased mortality being due to deferred hospitalizations. Second, it is possible that during these time periods a more virulent strain was circulating in Israel. However, there is no evidence for the existence of such a strain and the fact that the time period lasted for only 7 weeks, and was followed by a time period in which the model achieved accurate predictions makes it unlikely. The fact that increased mortality is once again observed in time-period IV, in which hospital load is once again high, further strengthens our hypothesis.

Our study has several limitations. First, the increased mortality observed during periods of high rate of COVID-19 related morbidity may be due to factors specific to the Israeli healthcare

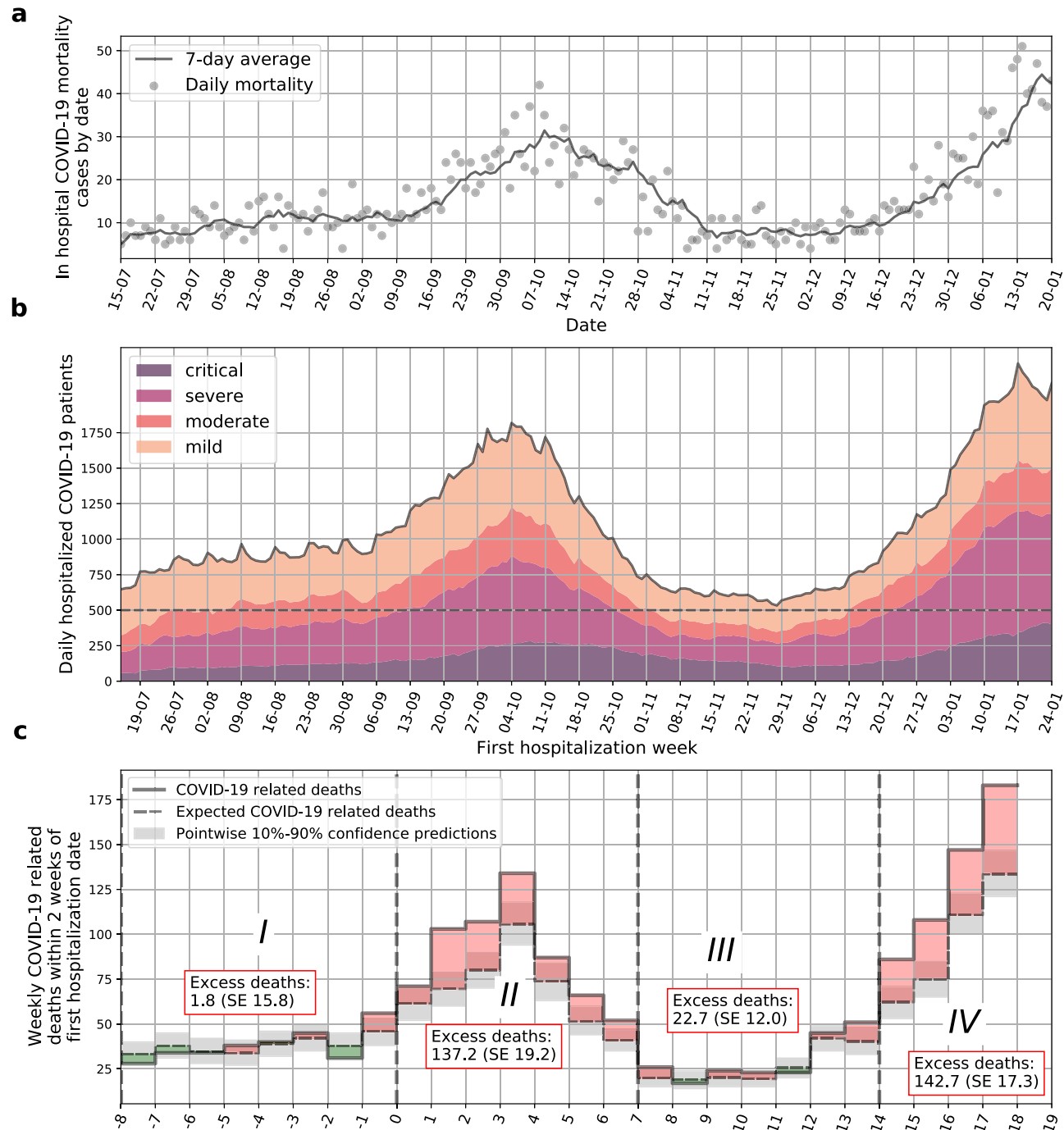

**Fig. 1 Hospital load and increased COVID-19 related mortality. a** In hospital COVID-19 related mortality cases by date. Daily mortality is marked by gray circles, and a 7-day average is plotted as a solid line. **b** Number of COVID-19-related hospitalizations by day in all hospitals in Israel. Colors depict different clinical states. A threshold of 500 severe and critical patients per day is presented by a dashed horizontal line. Dashed vertical lines separate weeks which are over/under this threshold. **c** COVID-19 related deaths for sets of patients that were first hospitalized in different weeks. Observed deaths for each week are marked by a solid line. Expected deaths as predicted from the model are marked in dashed black line. Monte-Carlo-based pointwise 10–90% confidence predictions are marked in gray. Weeks in which excess deaths were observed (positive difference between true and expected curves) are filled in red; and weeks in which deaths were overestimated (negative difference between true and expected curves) are filled in green. *X*-axis is marked by the week number. SE standard error.

system. The Israeli healthcare system is universal and mandates all citizens to join one of the official non-profit health insurance organizations. Regional[11] or financial[12] disparities affecting the availability of health-care resources in other countries as well as racial variation[13] may affect COVID-19-related mortality. Future studies should be conducted in order to determine if this effect is

also observed in other countries with different healthcare systems, and the specific threshold of patients representing healthcare capacity in each healthcare system. Full model code is available for use at https://github.com/JonathanSomer/covid-19-multi-state-model. Second, our findings may be influenced by the availability of diagnostic testing for COVID-19 as well as accurate

**Table 1 Cohort characteristics.**

| Characteristic | All | Time periods | | | |
| | | I | II | III | IV |
|---|---|---|---|---|---|
| Weeks | – | (−8) to (−1) | 0-6 | 7-13 | 14-18 |
| Date of first hospitalization | – | 15/07/2020–08/09/2020 | 09/09/2020–27/10/2020 | 28/10/2020–14/12/2020 | 15/12/2020–20/01/2021 |
| No. (%) | 22,636 (100.00) | 5966 (26.36) | 7418 (32.77) | 2761 (12.20) | 6491 (28.68) |
| Female, no. (%) | 11,070 (48.90) | 3082 (51.66) | 3491 (47.06) | 1362 (49.33) | 3135 (48.30) |
| Age, mean (SD) y | 59.79 (21.91) | 56.71 (22.12) | 59.71 (21.84) | 60.48 (21.18) | 62.44 (21.73) |
| Age, median y | 63 | 60 | 63.5 | 64 | 66 |
| Age group, no. (%) y: | | | | | |
| <20 (%) | 958 (4.23) | 311 (5.21) | 309 (4.17) | 110 (3.98) | 228 (3.51) |
| 20–29 (%) | 1620 (7.16) | 511 (8.57) | 541 (7.29) | 180 (6.52) | 388 (5.98) |
| 30–39 (%) | 1849 (8.17) | 540 (9.05) | 634 (8.55) | 206 (7.46) | 469 (7.23) |
| 40–49 (%) | 2308 (10.20) | 732 (12.27) | 748 (10.08) | 268 (9.71) | 560 (8.63) |
| 50–59 (%) | 3079 (13.60) | 869 (14.57) | 978 (13.18) | 400 (14.49) | 832 (12.82) |
| 60–69 (%) | 4142 (18.30) | 1096 (18.37) | 1391 (18.75) | 512 (18.54) | 1143 (17.61) |
| 70–79 (%) | 4097 (18.10) | 952 (15.96) | 1325 (17.86) | 549 (19.88) | 1271 (19.58) |
| 80+ (%) | 4583 (20.25) | 955 (16.01) | 1492 (20.11) | 536 (19.41) | 1600 (24.65) |
| Initial clinical state at hospitalization, no. (%): | | | | | |
| Mild (%) | 13,508 (59.67) | 3885 (65.12) | 4760 (64.17) | 1502 (54.40) | 3361 (51.78) |
| Moderate (%) | 3600 (15.90) | 1011 (16.95) | 1034 (13.94) | 452 (16.37) | 1103 (16.99) |
| Severe (%) | 4967 (21.94) | 976 (16.36) | 1441 (19.43) | 724 (26.22) | 1826 (28.13) |
| Critical (%) | 561 (2.48) | 94 (1.58) | 183 (2.47) | 83 (3.01) | 201 (3.10) |
| Deaths within the prediction period, from hospitalization date, no. (%): | | | | | |
| 7 days (%) | 1009 (4.46) | 165 (2.77) | 369 (4.97) | 111 (4.02) | 364 (5.61%) |
| 14 days (%) | 1994 (8.81) | 359 (6.02) | 715 (9.64) | 242 (8.76) | – |
| 21 days (%) | 2532 (11.19) | 503 (8.43) | 919 (12.39) | 323 (11.70) | – |
| 28 days (%) | 2786 (12.31) | 592 (9.92) | 1007 (13.58) | 371 (13.44) | – |

Time periods were divided by the maximum daily number of severe and critical patients during the week: in time-periods I and III <500, and in time-periods II and IV >500.

**Table 2 Expected versus actual cumulative deaths for 14 and 28 days from the first day of the week of hospitalization.**

| Time period | | Expected cumulative deaths | Actual cumulative deaths | Excess deaths (actual–expected) | Percentage of actual deaths |
|---|---|---|---|---|---|
| I | 14-day mortality | 304.2 (SE 15.8) | 306 | 1.8 (SE 15.8) | 0.6 (SE 5.1) |
| | 28-day mortality | 547.9 (SE 19.8) | 570 | 22.1 (SE 19.8) | 3.9 (SE 3.5) |
| II | 14-day mortality | 482.8 (SE 19.2) | 620 | 137.2 (SE 19.2) | 22.1 (SE 3.1) |
| | 28-day mortality | 841.7 (SE 23.9) | 998 | 156.3 (SE 23.9) | 15.7 (SE 2.4) |
| III | 14-day mortality | 186.3 (SE 12.0) | 209 | 22.7 (SE 12.0) | 10.8 (SE 5.7) |
| | 28-day mortality | 329.0 (SE 15.0) | 359 | 30.0 (SE 15.0) | 8.4 (SE 4.2) |
| IV | 14-day mortality | 381.3 (SE 17.3) | 524 | 142.7 (SE 17.3) | 27.2 (SE 3.3) |

SE standard error.

and complete documentation of the disease severity state by clinicians. However, testing policy and physicians documentation practices did not change significantly in Israel throughout the study period. Moreover, we only included data after 13 July 2020, in which uniform criteria for COVID-19 disease severity were applied by the MOH in all hospitals in Israel.

While tempting to do so, it is a highly challenging task to estimate any functional dose-response relation from these results and data, and it is beyond the scope of this work. Nonetheless, we can say with confidence that above a certain number of hospitalized severe or critical patients (~450), excess death seems bound to occur.

In conclusion, here we have shown that the mortality of hospitalized patients with COVID-19 in Israel was associated with health-care burden, reflected by the simultaneous number of hospitalized patients in a severe condition. Our work emphasizes that even in countries in which the healthcare system did not reach a specific point defined as insufficiency, the increase in hospital workload was associated with quality of care and patient mortality, ruling out factors related to change in the hospitalized population. In addition, our study highlights the importance of quantifying excess mortality in order to assess quality of care, and define an appropriate carrying capacity of severe patients in order to guide timely healthcare policies and allocate appropriate resources.

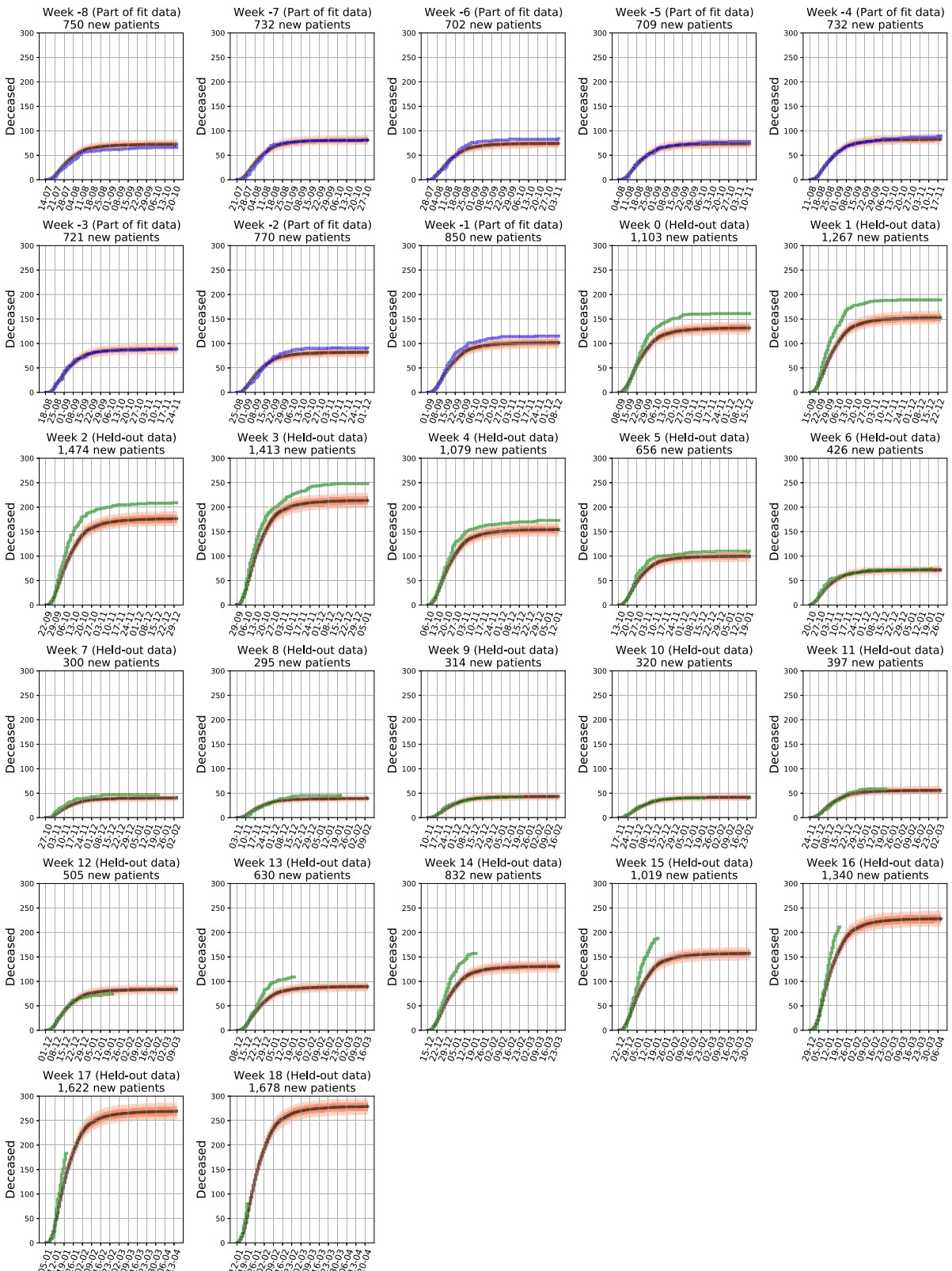

**Fig. 2 Cumulative expected and observed COVID-19 in-hospital deaths by week of first hospitalization.** Each sub-figure presents curves for the set of patients first hospitalized in a given week. Black curves are expected deaths with orange bands representing Monte-Carlo-based pointwise 10–90% confidence predictions. Blue and green curves are actual deaths for weekly sets of patients. Blue curves are drawn for patients which the model was trained on (weeks −8 to −1), and green curves are drawn for patients which the model was not trained on (weeks 0–18).

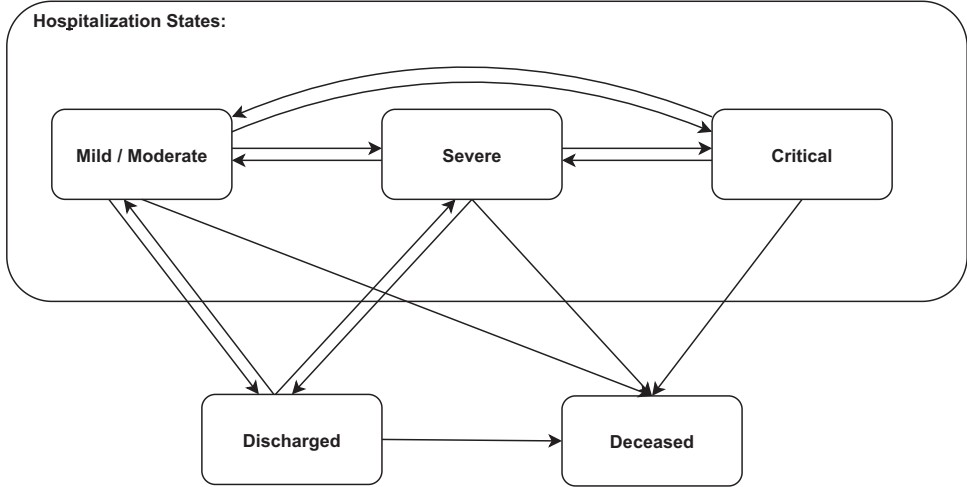

**Fig. 3 Multistate model.** Patients disease course transitions between 5 possible clinical states: mild or moderate, severe, critical, discharged, and deceased. Each transition was modeled using a set of Cox regression models, adjusting for right censoring, recurrent events, competing events, left truncation, and time-dependent covariates.

## Methods

**Data**. We analyzed data originating from the Israeli MOH on COVID-19 related mortality in Israel from 15/07/2020 to 20/01/2021. Patient data included information on age, sex, date of positive SARS-CoV-2 polymerase chain reaction (PCR) test, date of hospitalization, and clinical outcome (death or discharge from hospitalization) for each individual. In addition, daily information on disease severity during hospitalization was available. Classification of disease severity was based on the following clinical criteria, applied on 13 July 2020 by the Israeli MOH: mild illness – individuals who have any of the various signs and symptoms of COVID-19 (e.g., fever, cough, malaise, and loss of taste and smell); moderate illness – individuals who have evidence of pneumonia by a clinical assessment or imaging; severe illness – individuals who have respiratory rate >30 breaths per minute, SpO2 <93% on room air at sea level, or ratio of arterial partial pressure of oxygen to fraction of inspired oxygen (PaO2/FiO2) <300 mmHg, and ventilated/critical (denoted in this paper as Critical) – individuals with respiratory failure who require ventilation (invasive or non-invasive), multiorgan dysfunction or shock[14]. These criteria were determined based on NIH[15] and WHO[1] definitions.

**Statistical analysis**. In order to assess whether mortality of hospitalized patients with COVID-19 in Israel was associated with health-care burden we applied a multistate prediction model. The model is a modification of a Cox regression-based survival analysis model previously described in a study by Roimi et al.[6]. which predicts the clinical course of individual patients. The model adjusts for right censoring, recurrent events, competing events, left truncation, and time-dependent covariates. The original aim of the model was to allow timely allocation of sufficient healthcare resources and skilled medical professionals by medical centers. Weekly predictions of mortality and number of severe cases based on the model were presented and utilized by policy makers in Israel.

A hospitalized patient is in one of four clinical states: mild, moderate, severe, or critical; the exact definition of the states is detailed above. The multistate model has five states: (i) mild or moderate, (ii) severe, (iii) critical, (iv) discharged, and (v) deceased. This multistate model consists of 14 Cox regression models, one for each possible state-to-state transition, shown in Fig. 3. The 14 semiparametric models each includes a set of covariates, possibly with time-dependent covariates and different covariates for each model. Specifically, we took in age, sex, and state at hospitalization as baseline covariates. We also added time-dependent covariates encoding the hospitalization history of the patient: cumulative days in hospital and whether the patient had been in a severe or critical state before.

Making predictions based on our proposed multistate model requires estimating the absolute risks, also known as the cumulative incidence functions. The absolute risks involve estimating the probabilities of moving between states, the time to be spent at each state and integrating over all possible combinations between any possible triplet of entry state, exit state, and hospital length of stay. Since hospitalization consists of potentially multiple transitions between transient states (up to 14 transitions for a patient), the absolute risks have no tractable analytic forms. Thus, we performed Monte-Carlo (MC) sampling from the multistate model, in order to obtain consistent predictions for each individual patient and for the cohort. A detailed description of the MC sampling procedure is given in Roimi et al.[6]. Results of the Cox survival analysis are shown in Tables 1–4 of the Supplementary Information.

**Reporting summary**. Further information on research design is available in the Nature Research Reporting Summary linked to this article.

## Data availability
The data that support the findings of this study originates from the Israel minister of health. Restrictions apply to the availability of these data and so are not publicly available.

## Code availability
Analysis source code is available at:
https://github.com/tomer1812/covid19-israel-multi-state-hospitalization-model
https://doi.org/10.5281/zenodo.4567352
All analyses were performed using the statistical software R version 4.0.3, and Python version 3.6.
Model source code is available at:
https://github.com/JonathanSomer/covid-19-multi-state-model

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

## Acknowledgements

M.G. received support from the U.S.-Israel Binational Science Foundation (BSF, 2016126). We thank the following for their contributions to our efforts: Meir Bruhim, Strategic Planning, Israeli MOH; Avidan Cohen, Business Intelligence division, Israeli MOH; Linoy Vaknin-Alon, Business Intelligence division, Israeli MOH; and Dr. Danny Eytan, Rambam Health Care Campus and Technion - Israel Institute of Technology.

## Author contributions

H.R. and T.M. conceived the project, designed and conducted the analyses, interpreted the results, and wrote the manuscript; J.S., R.G., and A.B.A. contributed to the statistical data analysis and interpreted the results; S.S. interpreted the results and wrote the manuscript; E.S. designed the analyses, interpreted the results, wrote the manuscript, and supervised the project. U.S. and M.G. designed the analyses, interpreted the results, wrote the manuscript, supervised, and conceived the project.

## Competing interests

The authors declare no competing interests.
