## [Peer Review File · Nature Communications]

REVIEWER COMMENTS

Reviewer #1 (Remarks to the Author):

This study by Rossman et al. has investigated the relation of hospital load and COVID-19 mortality. This is a very important question of the current pandemic, but not easy to be addressed, and attracted a lot attention to understand this. By performing a nationwide study in Israel with high quality data, this study provided compelling evidence that overwhelming hospital load is associated with increased COVID-19 morbidity in hospitalized patients. In general, this is a clear and well-performed study, and here are a few questions:

1. The authors described "In addition, we note that the increase in observed mortality is despite the fact that throughout the pandemic, clinical experience in treatment of COVID-19 patients increased, along with a better understanding in pharmacologic (such as corticosteroids 10 and remdesivir 11) and non-pharmacologic (such as proning 12) treatment modalities that may be beneficial for the patients.". This is a very interesting point further supporting the important role of hospital load in COVID-19 mortality. However, do the authors have the data on the use of these medications and perform additional analysis?
2. For modeling, where are the key parameters originated? These information and the mathematical formulas should be provided in Methods.
3. The current format appears as a short communication, whereas this journal usually publishes full articles. It is recommended to remove the supplementary figures to the main manuscript. Fig. S1 can be integrated into Figure 1. A detailed Methods section should be provided.

Reviewer #2 (Remarks to the Author):

1. This report provides some new information about rates of in-hospital mortality for patients as COVID prevalence varies.
2. There is some speculation (and indirect evidence) that criteria for hospitalization change as COVID prevalence increases -- <https://twitter.com/ashishkjha/status/1333636841271078912?lang=en>. If this is true, and the change in threshold incorporates variables that are unobservable in these data, the result could be biased. One way to check for this is to see whether the results would be even greater when failing to adjust for clinical state at all. If that is the case, then there is reason to believe the results are in part due to a changing threshold for admission.
3. It is odd to a non-Israeli reader to do these calculations at the national level. Presumably, if pressure on hospitals affects COVID mortality, that should be happening at a hospital-specific level. Surely there was variation in COVID cases among hospitals within Israel. The results would be more compelling if this internal variation could be exploited.
3. Alternatively, or in addition, at the national level, it would be more compelling if there were some sort of "dose-response" relationship, as cases approach and exceed the threshold.

Reviewer #3 (Remarks to the Author):

Rossman and Meir performed model-based analysis, to assess excess mortality d/t COVID-19 in Israel during what is described as 2nd and 3rd outbreaks of the disease. The model was verified before the current analysis, and again on the 1st week included in the study.

In a well written report, solid methodology and well-defined limitations, the authors show that

even in the "non-overload" range defined by a state, there was increased in-hospital mortality d/t COVID-19, compared to the prediction delta in periods of lower activity of the pandemic.

This is an important message. I have no comments for the authors.

REVIEWER COMMENTS

Reviewer #1 (Remarks to the Author):

This study by Rossman et al. has investigated the relation of hospital load and COVID-19 mortality. This is a very important question of the current pandemic, but not easy to be addressed, and attracted a lot of attention to understand this. By performing a nationwide study in Israel with high quality data, this study provided compelling evidence that overwhelming hospital load is associated with increased COVID-19 morbidity in hospitalized patients. In general, this is a clear and well-performed study, and here are a few questions:

Comment 1.1:

The authors described “In addition, we note that the increase in observed mortality is despite the fact that throughout the pandemic, clinical experience in treatment of COVID-19 patients increased, along with a better understanding in pharmacologic (such as corticosteroids 10 and remdesivir 11) and non-pharmacologic (such as proning 12) treatment modalities that may be beneficial for the patients.”. This is a very interesting point further supporting the important role of hospital load in COVID-19 mortality. However, do the authors have the data on the use of these medications and perform additional analysis?

Response 1.1:

We thank the reviewer for this comment. While we do not have granular medication data available at patient or hospital level, we have discussed this point with medical policy makers and clinicians in Israel. National treatment guidelines for COVID-19 are based on NIH policy and are described in more detail in this document (available in Hebrew only):

<https://www.health.gov.il/Subjects/disease/corona/Documents/guide-treating-corona-patients.pdf>

In practice it has been made clear to us that each hospital has its own protocols and therefore an analysis of the effect of different treatment modalities on mortality on a nationwide level is not feasible. Nonetheless, any effect of better treatment modalities discovered throughout the pandemic, as clinical experience in treatment of COVID-19 patients increased, would most likely be toward a decreased mortality rate and can not be accounted for by the increased mortality observed by us. As we noted in the text, the *“increase in observed mortality is despite the fact that throughout the pandemic, clinical experience in treatment of COVID-19 patients increased”*.

Comment 1.2:

For modeling, where are the key parameters originated? These information and the mathematical formulas should be provided in Methods.

Response 1.2:

Thank you for this comment. The revised paper includes a methods section in the main text, and detailed information regarding all model parameters is now presented in Tables S1-S4 of the Supplementary Information. Explanations and detailed statistical theory regarding the construction of the models and the prediction procedure is described in Roimi et al. JAMIA 2021 (ref 6 in the paper); a detailed code repository with examples is available at:

<https://github.com/JonathanSomer/covid-19-multi-state-model-wave2>.

Comment 1.3:

The current format appears as a short communication, whereas this journal usually publishes full articles. It is recommended to remove the supplementary figures to the main manuscript. Fig. S1 can be integrated into Figure 1. A detailed Methods section should be provided.

Response 1.3:

Thank you for this suggestion. Pursuant to your comment, we have now moved figures S1 and S2 to the main manuscript, and have also edited the paper to follow the article format required by Nature Communications, including a detailed methods section.

Reviewer #2 (Remarks to the Author):

Comment 1.2:

This report provides some new information about rates of in-hospital mortality for patients as COVID prevalence varies.

Comment 2.2:

There is some speculation (and indirect evidence) that criteria for hospitalization change as COVID prevalence increases -- <https://twitter.com/ashishkjha/status/1333636841271078912?lang=en>. If this is true, and the change in threshold incorporates variables that are unobservable in these data, the result could be biased. One way to check for this is to see whether the results would be even greater when failing to adjust for clinical state at all. If that is the case, then there is reason to believe the results are in part due to a changing threshold for admission.

Response 2.2:

Thank you for this comment. First, we believe that changes in hospitalization rates should count as components of the *total effect* of hospital load on excess mortality, and would like to propose the following DAG (Directed Acyclic Graph; see [Tennant et.al. https://doi.org/10.1093/ije/dyaa213](https://doi.org/10.1093/ije/dyaa213)) to further discuss this. Shown in Fig R1 below is a DAG that depicts proposed causal connections between different variables of interest.

The DAG shows different mediating paths by which High hospital load (H) might impact excess mortality (E). Explicitly - the effect of H may be mediated by “Late admissions” (L) and also by a path going through Change (deterioration) in care (C). As we cannot truly estimate causal effects of different causes and paths with the data available to us, we modestly attempt to estimate and describe the *total effect* of H on E, which may be a complex combination of the different mediation paths. Adding to the complexity is also a possible improvement in Treatment options (T), which was also described in Response 1.1. Although ,we can confidently say that the effect of better treatment options (T) on change in care (C) should be a positive one (improvement in care), and thus would only have lessened the impact of H on E. Note, the DAG is far from being exhaustive and is only given to illustrate our perspective on what can be estimated with respect to the reviewers comment.

Second, our model is configured as a **multistate** Cox regression-based survival analysis. As such, the model crucially depends on the clinical state the patient is in (Fig 3); this cannot be disconnected from the modelling choice presented in the paper. Namely, in the current modeling scheme, we cannot drop the clinical state at hospitalization from the model as it is an integral part of the multistate modelling; in fact one of the reasons we chose this model is in order to be able to adjust for issues such as differing severity of incoming patients over time. We believe that adjusting for state at hospitalization and throughout the entire period of hospitalization captures some of the biases that may arise due to late admissions at different phases of the pandemic.

As pointed by the reviewer, case counts are prone to change with changing testing policies and different infected population dynamics (such as school outbreaks). We wish to point out that as such, our work does not attempt to predict new hospitalizations from those with positive test for COVID-19, but rather starts at the much more reliable point of hospitalization for incorporating important variables such as state at hospitalization, age, sex and history of hospitalizations.

Comment 2.3:

It is odd to a non-Israeli reader to do these calculations at the national level. Presumably, if pressure on hospitals affects COVID mortality, that should be happening at a hospital-specific level. Surely there was variation in COVID cases among hospitals within Israel. The results would be more compelling if this internal variation could be exploited.

Response 2.3:

We thank the reviewer for this comment and agree that a non-Israeli reader may not be aware of the fact that there were relatively small hospital-specific variations during the pandemic. Israel is a small and heavily urbanized country (over 90% of the population), with the vast majority of its population living in a narrow band of land. According to the Israel Central Bureau of Statistics, roughly 74% of the country's population live in one of the 4 big metropolitan areas (Tel-Aviv, Jerusalem, Haifa, Beer-Sheva), which are all less than 200km apart from each other.

As Israel has a relatively small geographical area and small population, and national restrictions were placed on the entire country at the same time (such as school closures and lockdowns) - cases and hospitalizations dynamics were similar across different hospitals, and so the analysis was conducted on a national level. We now emphasise this point in the main text as well.

This nation-wide similarity can be roughly seen in Fig R2 where new severe or critical hospitalizations are plotted for the country's 15 biggest hospitals (that account for ~75% of national COVID-19 related hospitalizations during this period). Overall, individual hospitals follow the national hospitalization trend (depicted in Fig 1.B) and do not show a substantial variation. Moreover, as many hospitals in Israel are in geographic proximity to each other, when a hospital reaches its hospitalization capacity, it can inform Pre-Hospital Medical Services which will redirect incoming patients to other hospitals nearby, effectively creating a national stabilization mechanism that minimizes variations in hospital load. This mechanism works even between different metropolitan centers, for example redirecting patients from the Jerusalem metro-area to hospitals in Tel-Aviv. We wish to add that due to restrictions imposed by the Israeli MOH, we are not authorized to report individual hospital statistics such as estimated excess deaths.

Fig R2. New hospitalizations in 15 of Israel's largest hospitals (7 days rolling average). The four time periods I-IV are marked and separated by dashed vertical lines as in Fig 1. (separating time periods of transition from over/under 500 national severe or critical hospitalizations). This figure is now also added as Fig S1 in the supplementary information.

Comment 2.4:

Alternatively, or in addition, at the national level, it would be more compelling if there were some sort of "dose-response" relationship, as cases approach and exceed the threshold.

Response 2.4:

We agree that it may be of interest to explore any possible dose-response relationship. Below in Fig. R3, we plot the weekly 14-day excess deaths percentage (of observed deaths) as a function of the 14-day average hospital load of severe and critical patients. Error bars mark the SE for the excess death percentage. We observe an interesting pattern where under ~450 hospitalizations (severe or critical), there are positive and negative excess deaths. However, above ~450 severe or critical hospitalizations, all weeks show pointwise significant excess deaths at rates of 10-30% of the true deaths for that weekly cohort. We believe it is a highly challenging task to estimate any functional dose-response relation from these results and data. Nonetheless, we can say with confidence that above a certain number of hospitalized severe or critical patients (~450), excess death seems bound to occur. We have now added this finding to the manuscripts discussion.

Fig. R3. Weekly 14-day excess deaths percentage (of true deaths) as a function of the 14-day average hospital load of severe and critical patients. Error bars mark the SE for the excess death percentage.

Reviewer #3 (Remarks to the Author):

Rossmann and Meir performed model-based analysis, to assess excess mortality d/t COVID-19 in Israel during what is described as 2nd and 3rd outbreaks of the disease. The model was verified before the current analysis, and again on the 1st week included in the study.

In a well written report, solid methodology and well-defined limitations, the authors show that even in the "non-overload" range defined by a state, there was increased in-hospital mortality d/t COVID-19, compared to the prediction delta in periods of lower activity of the pandemic.

This is an important message. I have no comments for the authors.

Response 3.1

Thank you!

REVIEWERS' COMMENTS

Reviewer #1 (Remarks to the Author):

Issues addressed.

Reviewer #2 (Remarks to the Author):

The authors have adequately responded to my concerns.